# Clinical and MRI-Based Assessment of Patients with Temporomandibular Disorders Treated by Controlled Mandibular Repositioning

**DOI:** 10.3390/diagnostics14060572

**Published:** 2024-03-07

**Authors:** Diwakar Singh, Alain Landry, Martina Schmid-Schwap, Eva Piehslinger, André Gahleitner, Jiang Chen, Xiaohui Rausch-Fan

**Affiliations:** 1Center for Clinical Research, University Clinic of Dentistry, Medical University of Vienna, 1090 Vienna, Austria; diwakar-divakarjit.singh@meduniwien.ac.at; 2Department of Education in Occlusion Medicine, Vienna School of Interdisciplinary Dentistry (VieSID), 3400 Klosterneuburg, Austria; i.d.alainlandry@gmail.com; 3Division of Prosthodontics, University Clinic of Dentistry, Medical University of Vienna, 1090 Wien, Austria; martina.schmid-schwap@meduniwien.ac.at (M.S.-S.); eva.piehslinger@meduniwien.ac.at (E.P.); 4Division of Radiology, University Clinic of Dentistry, Medical University of Vienna, 1090 Vienna, Austria; andre.gahleitner@meduniwien.ac.at; 5School and Hospital of Stomatology, Fujian Medical University China, Fuzhou 350122, China

**Keywords:** temporomandibular disorder, Controlled Mandibular Repositioning, disc displacement with reduction, MRI, condylography

## Abstract

Background: Occlusal splints and anterior repositioning splints (ARSs) are widely accepted treatments for temporomandibular disorders (TMDs). However, there is uncertainty with regard to the most suitable amount of mandibular repositioning. The aim of this study is to evaluate the clinical and functional effects of the therapeutic position (ThP) established based on the Controlled Mandibular Repositioning (CMR) method. Methods: In this clinical trial, 20 subjects with 37 joints with disc displacement with reduction were recruited. The initial standard functional diagnostic protocol, MRI, and digital condylography were performed, and ThP was calculated with the CMR method. After a 6-month follow-up, the standard diagnostic protocol was repeated. The change in disc position was evaluated by means of MRI after 6 months of CMR therapy. Results: The MRI findings in the parasagittal plane demonstrated that out of the 37 joints presenting disc displacement, 36 discs were successfully repositioned; thus, the condyle–disc–fossa relationship was re-established. Therefore, the success rate of this pilot study was 97.3%. The mean position of the displaced discs was at 10:30 o’clock of the TMJ joint and at 12:00 o’clock after CMR therapy. Conclusions: The ThP determined using the CMR approach reduced all of the anteriorly displaced discs (except one). The CMR method allowed to define an optimum ThP of the mandible thus supporting patients’ effective adaptation to treatment position.

## 1. Introduction

According to the National Institute of Dental and Craniofacial Research (NIDCRN) of the National Institute of Health (NIH), temporomandibular disorders (TMDs) are a group of more than 30 conditions that cause pain and dysfunction in the jaw joints and muscles. The NIH have reported that about 11 to 12 million adults in the United States have pain in the region of the temporomandibular joints (TMJs), and in 2013, estimated the annual TMD management cost in the USA being up to USD 4 billion dollars. Based on the Diagnostic Criteria for Temporomandibular Disorders (DC/TMD) [1], the 12 common TMD symptoms include arthralgia, myalgia, local myalgia, myofascial pain, myofascial pain with referral, four disc displacement disorders, degenerative joint disease, subluxation, and headache attributed to TMD [1]. Studies have reported that between 60 and 70% of the general population has at least once described a sign of a temporomandibular disorder [2]. According to the (DC/TMD), internal derangements can be distinguished into four basic types of articular disc displacements: disc displacement with reduction, disc displacement with reduction with intermittent locking, disc displacement without reduction with limited opening, and disc displacement without reduction without limited opening [1]. Epidemiological study have reported an incidence of internal derangement in patients with TMD of up to 36.8% [3].

The DC/TMD recommend Magnetic Resonance Imaging (MRI) for the confirmation of a provisional clinical diagnosis [1]. MRI is considered the most effective and non-invasive imaging tool for the evaluation of TMJ soft tissues, the disc–condyle relationship, and for the determination of the character of disc displacement, including position and shape [3,4]. High-resolution MRI 3 tesla allows for a better visualization of bony alterations and joint effusions. Previous studies suggested that 3.0 T MRI at the same resolution as 1.5 T yielded better results for the perceptibility of joint structures without increasing the examination time. The 3 T unit delivers images of better quality regarding the evaluability of disc position and shape as compared to 1.5 T MRI and thus provides added diagnostic assurance that is critical for therapeutic decisions [5,6]. The standard protocol for an MRI diagnosis of anterior disc displacement uses the most superior surface (12 o’clock position) of the condyle as a reference point for the posterior band of the disc. The posterior band of the disk located anterior to the 12 o’clock position correlates to anterior disk displacement [7]. A few studies have shown that the average position of the posterior band in asymptomatic individuals varies from 5° to 10° from the 12-o’clock depending on the intermediate zone criteria [8,9,10,11].

MRI provides a better visibility of joint morphology in image form, as sometimes clinical diagnosis does not provide a clear picture [12,13]. Several studies have compared clinical TMD diagnosis and MRI findings of TMJs, and different results have been obtained, and the agreement between specific clinical diagnosis and MRI findings has been fair to poor [14,15,16]. In particular, it is not clear whether the change in disc position observed upon MRI correlates with the dynamic findings of the condylar movements during function.

Condylography is better suited for determining dysfunctional conditions in the dynamics of condylar movements. Thus, a combination of both methods, using MRI and condylography, seems appropriate for an improved understanding of the functional disturbances of the stomatognathic system [13,17,18,19,20].

The management options for TMDs mainly begin with conservative, non-surgical treatment [21,22]. The conservative treatment approach mainly consists of different types of occlusal splints, psychological counseling therapy, physiotherapy, oral or injectable pharmacotherapy, low-level laser therapy and heat therapy, etc.

Occlusal splints are one of the most widely accepted treatment choices for conservative therapy [23]. In the literature, there is a dilemma about the most suitable type of splint therapy among the different types of splint therapy [24]. The anterior repositioning splint (ARS), hard stabilization splint, soft stabilization splint, mini-anterior splint, and prefabricated splint approaches reportedly show varying degrees of clinical efficacy in the treatment of TMDs compared to no treatment [22,25]. Previous studies in patients with internal derangements have revealed that splints helped in achieving a statistically significant reduction in TMJ pain, TMJ noises and disability, with corresponding improvements in jaw function after the disc was repositioned [26,27]. Nonetheless, the exact mechanism of action of oral splints remains controversial [26,28,29]. At present, there is no consensus on the most suitable design for occlusal splints and the superiority of a particular splint over others [22,30].

In clinical practice, ARSs are the most popular choice for ADDwR patients [31]. It has been shown that anterior repositioning appliances can be fabricated both for maxillary or mandibular dental arches with full or partial coverage. Some studies have compared the position of articular disc in MRI with anterior repositioning splints, and the results reported in different studies vary, but there is a positive correlation showing that ARSs are relatively effective in disc displacement with reduction [32,33,34,35,36]. Once the disc is recaptured, then the disc–condyle complexes are subsequently ‘walked back’ along the posterior slope of the articular eminence by periodic modification, i.e., by grinding of the splint [37,38,39]. In a recent study, this walking back of the condyle was termed step-back anterior repositioning splint [40]. A long-term clinical trial with MRI evaluation demonstrated that mandibular repositioning can be effective in reducing disk displacement, particularly on anteriorly displaced disks. Anterior mandibular repositioning appears much less effective in cases where there is a transversal component to the disc displacement (medial or lateral disc displacement) [41]. However, there is still uncertainty about the most suitable amount of repositioning in cases with ADDwR. Some authors have suggested that, clinically, the edge-to-edge positioning of the incisors is the best for the fabrication of repositioning splints [35,36,42]. The range of clinical mandibular repositioning in previous studies varies from a minimum amount of repositioning, to 2 mm of repositioning, and then to the maximum amount of repositioning (edge-to edge-positioning of the front teeth) [37,42,43,44,45]. In all the mentioned above studies, however, the repositioning was arbitrary due to a lack of 3D operator control; there was no quantitative assessment of disc position after mandibular repositioning, and the success of the treatment was based on the resolution of symptoms rather than disc recapture.

The aim of this study was to define a ThP suitable for recapturing anteriorly displaced discs with reduction using the Controlled Mandibular Repositioning (CMR) method. ThP determination on a Condylar Position Variator (CPV) was based on 3D cartesian coordinates obtained from condylography. The assessment of the effect of therapeutic mandibular position on the position of the disc with respect to the condyle was performed by means of MRI diagnosis.

## 2. Materials and Methods

### 2.1. Selection of Patients

Ethics approval was obtained from the Medical University of Vienna, Austria (EKNr: 2267/2018). Subjects were recruited from the Special Clinic for Temporomandibular Disorders, Clinical Division of Prosthodontics, University Clinic of Dentistry, Medical University of Vienna. A required sample size of 16 patients was calculated, assuming an average difference in disc position before and after CMR therapy of 0.5 mm ± 0.5 mm. The calculation was conducted for a paired-sample *t*-test assuming a statistical power of 95% and a two-sided significance level of 5%. To compensate for potential dropouts in the course of the study, 20% (16 × 0.2 = 3.2 ≈ 4) more patients were included. The total sample size was therefore set to 20 participants. A more detailed description of the sample is shown in Table 1.

Inclusion criteria: Patients were enrolled based on the following inclusion criteria: (a) age between 18 and 45 years, (b) absence of any systemic diseases, (c) clinical diagnosis of ADDwR based on DC/TMD [1], (d) MRI confirmation of ADDwR in at least one joint (e) without arthrotic changes in medium or severe entity, and (f) ability to provide signed informed consent.

Exclusion criteria: The exclusion criteria was as follows: (a) pregnancy; (b) congenital abnormalities or dentofacial deformities; (c) recent oro-facial surgery, cervical trauma, or major accidents; (d) major psychological disorders; (e) complete or partial dentures; (f) prior TMD treatment; (g) two or more teeth missing in one quadrant; (h) periodontal problems; and (i) claustrophobia.

Out of the 20 subjects, 17 patients presented bilateral ADDwR, and the remaining three patients presented with ADDwR on one side and ADDw/oR on the other side. All the subjects enrolled in the study had to undergo the standard anamnesis, which included detailed records of each subject’s medical and dental history, the INFORM DC-TMD (https://inform-iadr.com/ accessed on 29 February 2024) protocol evaluation of joint clicking, acquisition of dental impression and the fabrication of split dental casts, TMJ MRI, palpation of muscles and TMJ structures, visual analogue scale (VAS, Figure 1), and condylography.

### 2.2. Clinical Examinations

#### 2.2.1. Sensitivity to Palpation of Muscles and Ligaments

The following 13 muscles and ligaments involved in TMJ function on the left and right sides were palpated: tuber maxillae, medial pterygoids, mylohyoids, superficial masseters, temporalis tendons, digastrics, TMJ lateral poles, deep masseters, temporomandibular ligaments, retral joint spaces, omohyoids, upper trapezius, and posterior temporalis. The palpation of these structures implied proper localization of the sites and an amount of pressure ranging from 200–500 g, which is dictated by the size and the firmness of the structure. The VAS was used for evaluation by the patients and the operator. There are two sides of the scale: one is the patient side (Figure 1a), with colors ranging from white to dark red, where a red color signals greater pain. The other side, i.e., the operator side (Figure 1b), shows numbers from 0 to 10, representing the intensity of pain.

**Figure 1 diagnostics-14-00572-f001:**
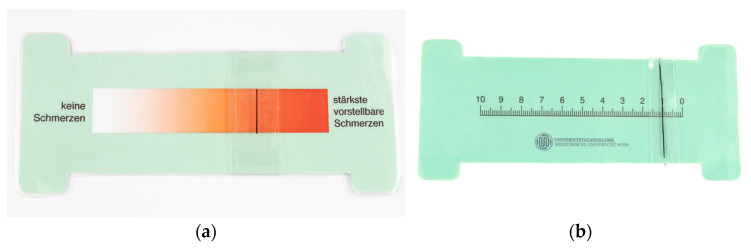
Visual analogue scale in use at the University Clinic of Dentistry, Medical University of Vienna: (**a**) patient side and (**b**) operator side of the scale.

#### 2.2.2. Condylography

Condylography is a method for monitoring condylar motion in 3D and is part of the clinical instrumental analysis for functional diagnosis (S2K guidelines German Society of Craniomandibular Function and Disorders (DGFDT https://www.quintessence-publishing.com/downloads/cmf_2023_03_s2k_guideline.pdf accessed on 29 February 2024). For this study, it was performed using CADIAX 4 (GAMMA, Klosterneuburg, Austria, https://www.gammadental.com/en/ accessed on 29 February 2024, 20:00). Condylography recordings were performed before and after CMR therapy after the identification of the true hinge axis, and the tracings reflected the movements across three planes with reference to the axis–orbital plane. The X-axis is antero-posterior, while Y and Z represent the transverse and vertical axes, respectively. Based on the patient’s needs, different types of standard and functional movements can be recorded. Condylography tracings (Figure 2a) show how the condyles, and therefore the mandible, are moving across all 3 planes. The starting point of all the movements was reference position (RP). RP is the retral border position of the mandible, in which the joint structures are not stressed (Rudolf Slavicek, The Masticatory Organ, 2002).

Figure 2b illustrates clicking during condylographic tracing. Figure 2c shows the coordinates for each joint separately at a particular selected point of movement from the condylographic tracing. After condylography, patients’ upper and lower casts could be accurately transferred to an articulator.

#### 2.2.3. Magnetic Resonance Imaging

The recruited patients were subjected to MRI examinations at the Clinical Division of Radiology, University Clinic of Dentistry, Medical University of Vienna. MRI was performed in maximal intercuspal position (ICP) (Figure 3a) and in open mouth position (Figure 3b). Instructions were given to patients by a senior radiology assistant. These instructions included the diligent maintenance of body position. Head movements were limited by restriction pads. The Magnetom Skyra (Siemens, Erlangen, Germany) MRI device with a field strength of 3 T and 16-channel head–neck coil was used. Scans of 5 min and 30 s for sagittal and coronal slices were performed; proton-weighted TSE images (TR 2300 ms, TE 10 ms, flip angle 160°, averages 2, concatenation 1, band width 300 Hz/Px, distance factor 10%, image resolution 0.3 × 0.3 × 2 mm voxels, field of view 17 cm) were produced. A transversal localizer scan was used for the detection of condyle position, and paracoronal slices were aligned to the axis of the condylar head. Sagittal slices of 1.1 mm were adjusted at right angles to the condylar head long axis and parallel to the mandibular ramus.

The position of the discs was evaluated based on the clock position according to Katzberg & Tallents [46]. In this approach, the image of the TMJ visualized in parasagittal view is superimposed to a clock. The disc position is considered normal when the posterior band of the articular disc was above the condylar head (12 o’clock). The steepness of the articular eminence was also considered, which shows that, sometimes, the 11:30 position can also be normal in steep articular eminence cases [47]. Figure 3b shows an anteriorly displaced articular disc at the 10 o’clock position.

After 6 months of CMR therapy, a control MRI examination of both TMJs were performed with the same technical MRI specifications but with the CMR stabilizer in place. The MRI evaluations were carried out by the specialist of the (A.G) clinical division of radiology and the specialist (M.S-S) of Special Clinic for Temporomandibular Disorders, who have over 25 years of clinical experience in the field of TMD. Both specialists (A.G and M.S-S) jointly analyzed the position of the articular discs before and after CMR therapy. Seven slices per joint for 37 joints with slice thicknesses of 1.1 mm were selected, so that the position of the articular disc could be seen clearly between the eminence and the condylar fossa. An anteriorly displaced disc at the 10 o’clock position before MRI treatment is shown in Figure 3a and Figure 4a. The articular disc was reduced upon MRI with the CMR stabilizer in the mouth (Figure 4b), and the posterior band of the disc was in the 12 o’clock position (Figure 4c).

### 2.3. Controlled Mandibular Repositioning

The Controlled Mandibular Repositioning (CMR) method was developed by Dr. Alain Landry in 1994. After a thorough clinical examination and diagnosis, this method of initial therapy consists of finding a ThP for the mandibular condyles before making a CMR stabilizer. It is based on cephalometric and condylographic tracing analysis.

The position of the maxillary cast is registered by the mean of a kinematic facebow transfer stand. The mandibular cast is mounted in the CPV (Figure 5a) (Gamma dental Klosterneuburg, Austria) with a reference position (RP) bite. The CPV allowed us to reposition each joint separately in all 3 planes of space and individually based on the cartesian coordinate system (X, Y, Z axes) with an accuracy ≤ 0.1 mm (www.gammadental.com accessed on 29 February 2024, 20:00). The changes brought to the condylar position were based upon the subject’s individual condylographic tracings and their sensitivity to palpation of muscles and ligaments.

Based on condylographic data, adjustments were made on a CPV at the condylar level; then, a bite registration in therapeutic position (test bite) was fabricated. After setting and trimming, this test bite was inserted into the patient’s mouth, and the patient was asked to open and close and to protrude and retrude to clinically verify whether the luxation was still present. If the luxation was still present, the same process was applied again with adjusted parameters given by condylographic tracings until there was no more detectable joint clicking. Once there was no more clicking, the patient was asked to bite on the test bite for 4 min, after which the palpation of muscles and ligaments was performed again. Based on the reduction in the sensitivity to muscle and ligament palpation (VAS), the test bite was refined. The position achieved with the final test bite corresponded to ThP. From this therapeutic position, a full-coverage mandibular CMR stabilizer (Figure 5c) was fabricated using self-cure translucent dental orthodontic resin material (Scheu dental, Germany, https://scheu-dental.com/fileadmin/six/4390313_E-STEADY-RESIN-Polymer_Vario.pdf accessed on 29 February 2024, 20:30). After insertion in the mouth, the proper fit of the CMR stabilizer on mandibular teeth was verified as well as the occlusion parameters (Figure 5d).

### 2.4. Clinical Follow-Up Evaluation

The patients were informed to wear their CMR stabilizer day and night, except during mastication and cleaning. The subjects were recalled every month for 6 months for follow-up appointments. After 6 months, the subjects were assessed clinically for all the signs and symptoms based on the DC/TMD and joint auscultation with their CMR stabilizer in their mouth. A control subjective pain/discomfort evaluation was performed using the VAS. MRI with the CMR stabilizer in the mouth was performed for all the subjects for an assessment of the position of the articular discs after 6 months of CMR therapy.

### 2.5. Statistical Analysis

Statistical analyses were conducted using IBM^®^ SPSS^®^ Statistics, version 29, and R Statistics, version 4.2.1. Mean and standard deviation, as well as minimum and maximum values, were calculated for disc displacement in clock position (from Katzberg) [46], separated by time points, by joints on the parasagittal plane. For each slice and each joint, a paired *t*-test was performed to determine whether the difference in clock position before and after CMR therapy was statistically significant. Bonferroni correction was applied due to multiple testing. A significance level of 5% was used. The means were plotted using the standard error of the mean to display the error bars. Means and standard deviations were calculated for the outcomes of the VAS test for each anatomical structure in different joint positions (in RP, ICP, and ThP) for both sides. A linear mixed model was conducted to analyze the average differences between the sensitivity to muscle and ligament palpation in different positions (i.e., RP/ICP/ThP), measured with the VAS. Patient ID and anatomical structure types were introduced into the model with random intercepts. Bar charts were used to plot the mean values, with the standard errors of the mean representing the error bars.

## 3. Results

Clinical evaluation and VAS: Out of the 37 included joints, in 36 joints, the discs were recaptured, as there was no sign of luxation or clicking based on the clinical examination and the MRI. Only one patient with both sides of ADDwR complained about occasional clicking and mild pain in one joint when she woke up in the morning. This was then confirmed by a control MRI.

The sensitivity of muscles and ligaments showed a significant improvement in pain on the VAS. The differences between the sensitivity to palpation in the RP, ICP, ThP positions are shown in Table 2 and displayed in Figure 6.

The sensitivity to pain in muscle palpation was significantly reduced in every structure palpated (Figure 6a–m). The tuber maxillae, medial pterygoids, superficial masseter, temporalis tendon, deep masseter, and temporomandibular ligaments were sensitive in RP and ICP. Their sensitivity was close to zero after 4 min in ThP (Figure 6a,b,d,e,h,i). The postural omohyoid and upper trapezius muscles (Figure 6k,l) saw their sensitivity reduce.

In case no. 9, where the patient still complained of intermittent clicking after using the splint, a second condylography was performed with the stabilizer in the mouth. A second session of CMR was performed, and a 0.4 mm repositioning of the affected joint was carried out. The occlusal part of the stabilizer was modified to make it coincide with the new ThP. A second control MRI was performed, which confirmed the reduction of the disc in the new ThP.

MRI evaluation: The type of internal derangement is shown in Table 3 (right versus left side). Anterior, antero-lateral, and antero-medial displacements were observed, while no medial, lateral, or posterior disc displacements were diagnosed.

Our MRI evaluations showed a statistically significant difference between the two measurement points (before therapy and after 6 months of CMR therapy) regarding the mean of disc displacement in clock position for each slice and for both joints (*p* < 0.001, Table 4). After the CMR therapy, the mean clock positions were close to 12 o’clock for all slices and for both joints. The distance between the mean value and 12 o’clock, which is the normal position, was less than one standard error of the mean in all cases. Before CMR therapy, the mean values were close to 10:30. The highest mean value was 10:32 with respect to the right joint (Figure 7a) and 10:36 regarding the left joint (Figure 7b). After CMR therapy, no subject had position values which were smaller than 11:30 or larger than 13 o’clock. Before CMR therapy, the individual values ranged between 9:30 and 11:30; thus, no subject had a normal position (see also Table 4).

## 4. Discussion

Condylography provides a detailed analysis of movements in terms of quantity, quality, characteristics, symmetry, reproducibility, and special findings [13,17,19]. MRI is a recognized and effective method for the diagnosis of internal derangements. For the treatment of internal derangements, there is also a need for functional diagnosis, including muscle status and condylar movement evaluation. Previous studies have compared the sensitivity and specificity of computerized condylography with magnetic resonance imaging, and the sensitivity varies from 40% to 86%, while the specificity varies from 90 to 100% [13,17,19]. 

Two main theories on ARS exist pertaining to their usefulness in reducing TMJ pain, clicking, and dysfunction. One theory asserts that ARSs allow for the displaced discs to slip back into their normal positions through a new mandibular position, which has been shown to be the edge-to-edge position in few studies [37,38,39]. Another theory proposes that ARSs reposition condyles anteriorly to catch or ‘re-capture’ displaced discs, establishing normal disc–condyle relationships within the glenoid fossae.

Different positions of splints have been estimated; for example, the edge-to-edge position, the 2 mm protrusive position, and the step-back anterior reposition. Repositioning was performed mostly arbitrarily in most of the studies [35,36,42,48] which showed that the overall success rate of anterior repositioning splints ranged between 50.0% and 70.0% based on clinical evaluation. In Tecco et al. study [49], when controlled with MRIs, the initial percentage of success was increased up to 90.0% with the teeth in an edge-to-edge position, but only 54.8% of the displaced discs were able to maintain appropriate disc–condyle connections in their least-protruded position. In another study, ARSs were created with incisors in the edge-to-edge position. The success rate rose to 93.5%, which was reduced later on due to grinding of the anterior repositioning splints during the follow-up appointments [36]. 

This methodology was called the “walking back of condyles”, as the edge-to-edge position was judged to be too far forward and too demanding for the stomatognathic system. During the follow-up appointments, the grinding of the splints to favor a more retral position of the mandible often resulted in the re-occurrence of joint luxations. Therefore, the initial reduction in luxations was only temporary [38,49,50].

The specific aim of the CMR method is to find a therapeutic position to reduce joint luxations based on the individual condylography 3D data and then refine the therapeutic position by reducing the sensitivity to palpation of the muscles and ligaments related to the stomatognathic system. The position of articular discs was evaluated before treatment and after 6 months of wearing a CMR stabilizer. In our MRI evaluations, of the 37 joints with disc displacements, in 36 joints, the articular disc was recaptured with a CMR stabilizer. The success rate of this treatment initially reached 97.3%.

The final outcome, if we include the correction brought to the joint where clicking re-appeared, brings the success rate of disc repositioning to 100%. MRI, the patient’s condylographic data, and the CMR method play a very important role in identifying ThP For reducible joint luxations, the coordinates of condylar position are chosen on the incursive tracings right before the luxation re-occurs. This helps to determine the antero-inferior repositioning of the condyles.

In contrast to ARSs and stabilizing splints, the design of a semi-anatomic hard full-coverage CMR mandibular stabilizer allows the antagonist teeth to sit in the indentations of the CMR stabilizer in ThP. Posterior indentations, canine guidance, and retrusive guidance are integrated into the design of CMR stabilizers, which allow for normal functions and functional freedom.

A strength of our study was that during the control appointments, there was no “walking back” of the condyles. There was no need for the step-back anterior repositioning splint retraction method [40], as, in our study, the ThP was calculated individually by the CMR method. Stabilizer adjustments were only made to maintain the final therapeutic position. In our follow-up appointments, we paid attention to the retrusive guidance so that there would not be a minute loss of the ThP given by the CMR stabilizers. The occlusion of the CMR stabilizers was verified for good occlusal support on each tooth from the first premolar to last molar, along with canine guidance providing posterior disocclusion during function. In this study, the therapeutic position increased the posterosuperior joint space, which unloaded the joints, and this finding is quite similar to those found in previous studies [51]. The results of the present study in relation to TMJ clicking are similar to those of past studies [52,53].

After 6 months, the VAS scores demonstrated the high efficiency of the CMR stabilizer treatment in terms of improving clinical symptoms. The intensity of pain was significantly reduced in this study, similar to previous studies [54,55]. It is necessary to emphasize that some subjects with postural problems were recommended for interdisciplinary treatment (for example, adjuvant physiotherapy) [56].

The limitations of this pilot case–control study include the fact that a small number of patients were recruited and the fact that the follow-up lasted only 6 months. A future study with a larger number of participants is required with a longer follow-up.

## 5. Conclusions

The key outcome of this pilot study is that in patients with disc displacement with reduction, using the CMR method resulted in 97.3% of the discs being repositioned to their normal position. The results also show high efficiency in terms of reducing the sensitivity to palpation of the muscles and ligaments of the stomatognathic system. The treatment identifying the optimal ThP of the condyles based on the CMR method allowed for recapturing displaced discs with reduction using a conservative amount of antero-inferior condylar repositioning and resulted in an appreciable decrease of patients’ symptoms.

## Figures and Tables

**Figure 2 diagnostics-14-00572-f002:**
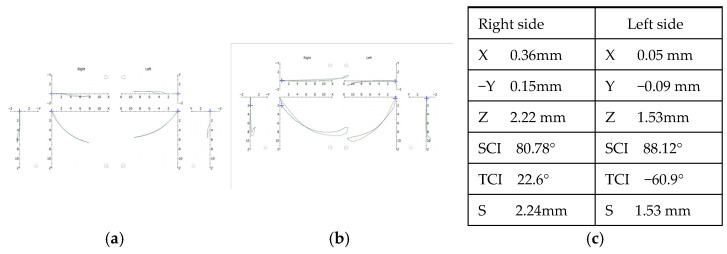
Condylographic data: (**a**) normal condylographic tracing without clicking; (**b**) protrusion retrusion with clicking; (**c**) X, Y, Z coordinates of left right and left joint.

**Figure 3 diagnostics-14-00572-f003:**
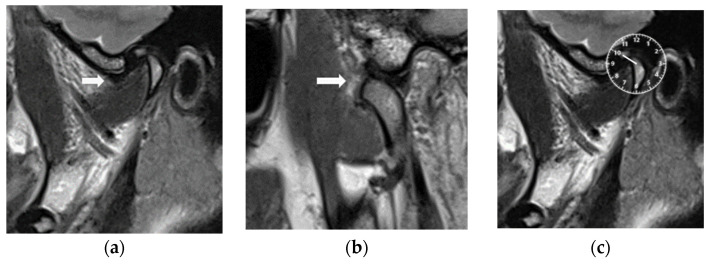
T2 images of magnetic resonance imaging (MRI): (**a**) maximal intercuspal position (white arrow shows the position of displaced disc); (**b**) open mouth position (white arrow shows the position of disc above the condylar head). (**c**) The disc is displaced in the 10 o’clock position based on Katzberg & Tallents [46].

**Figure 4 diagnostics-14-00572-f004:**
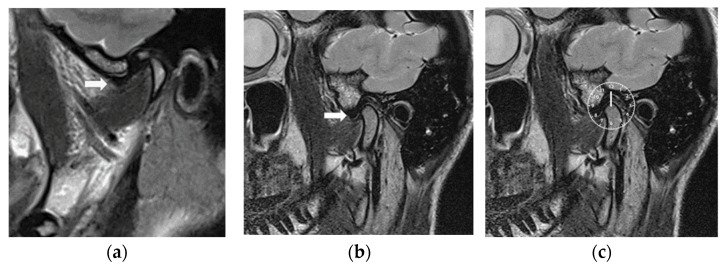
(**a**) Initial MRI (white arrow shows the disc displaced anteriorly before treatment). (**b**) Control MRI with CMR stabilizer in the mouth and white arrow showing the correct position of the disc. (**c**) Disc in the 12 o’clock position with CMR stabilizer in mouth.

**Figure 5 diagnostics-14-00572-f005:**
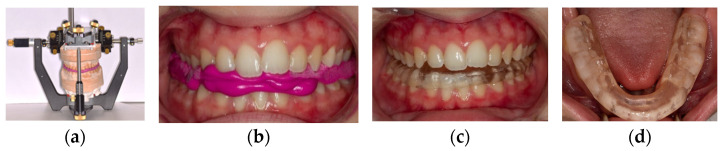
Fabrication of test bite and CMR stabilizer in therapeutic position: (**a**) test bite in CPV; (**b**) test bite in the mouth; (**c**) CMR stabilizer; (**d**) occlusal view of the CMR stabilizer.

**Figure 6 diagnostics-14-00572-f006:**
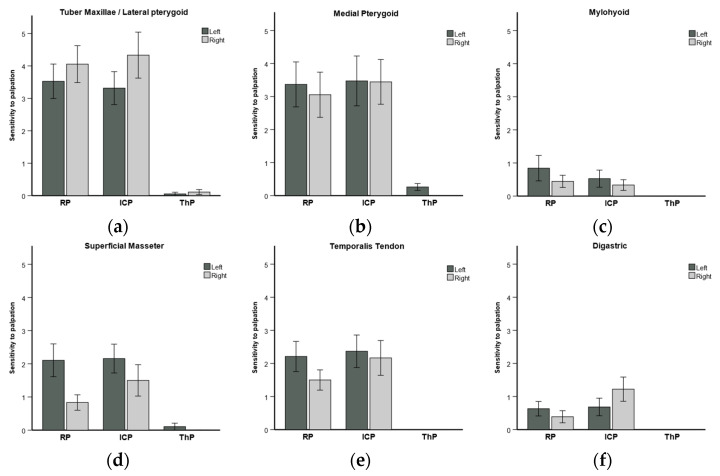
Sensitivity to palpation of pain in muscles and ligaments assessed using the visual analogue scale (VAS) in reference position (RP), maximum intercuspal position (ICP), and therapeutic position (ThP).

**Figure 7 diagnostics-14-00572-f007:**
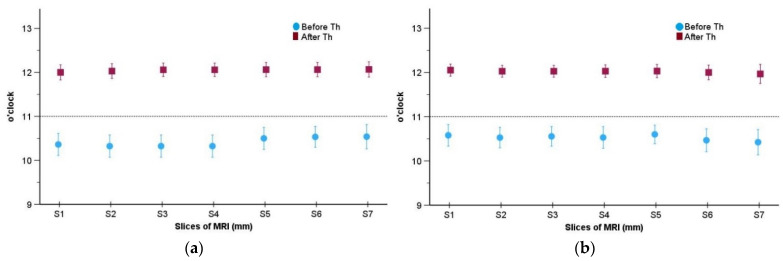
Graphical demonstration of the mean and standard deviation of disc displacement in clock position before and after CMR therapy for seven slices on the parasagittal planes for (**a**) right joint and (**b**) left joint (error bars represent standard error of the mean).

**Table 1 diagnostics-14-00572-t001:** Patients’ distribution.

No. Patients	Females	Males	Age in Years Mean ± SD Range	Joints with ADDwR	Excluded Joints
20	18	2	28.7 ± 6.4 (18–41)	37	3 *

* These joints presented ADDw/oR (non-reducible joint luxation).

**Table 2 diagnostics-14-00572-t002:** Visual Analogue scale differences between reference position, intercuspal position, and therapeutic position.

Parameter	β	SE (β)	df	t	*p*-Value	95% CI
RP vs. ThP	1.325	0.079	1302	16.721	<0.001 *	[1.170; 1.480]
ICP vs. ThP	1.517	0.079	1302	19.147	<0.001 *	[1.362; 1.673]
ThP (Intercept)	0.081	0.088	479.2	0.921	0.358	[−0.092; 0.253]

Outcome = VAS (0–10); β = regression slope; SE (β) = standard error for β; df = degrees of freedom; CI = confidence interval for β. Linear mixed model with random intercepts for patients and for muscle type. * Average pain significantly differed from average pain in the ThP position. VAS: 0 = ‘no pain’; 10 = ‘severe pain’; 1.8 ± 1.0; range = 0–10.

**Table 3 diagnostics-14-00572-t003:** Distribution of internal derangement: right versus left side.

	Right Joint	Left Joint
Anterior disc displacement	5	6
Antero-lateral disc displacement	12	12
Antero-medial disc displacement	1	1

There were no true medial, lateral, or posterior disc displacements in the study patients.

**Table 4 diagnostics-14-00572-t004:** Disc displacement in clock position before and after CMR therapy for seven parasagittal planes for right and left joints.

Joint	MRI Slices	Before		After		
		N	Mean ± SD	N	Mean ± SD	*p* Value
Right	1	18	10:21 ± 0:30	18	12:00 ± 0:20	0.0002
	2	17	10:19 ± 0:29	18	12:01 ± 0:19	0.0008
	3	17	10:19 ± 0:29	17	12:03 ± 0:18	0.0006
	4	17	10:19 ± 0:29	17	12:03 ± 0:18	0.0006
	5	15	10:30 ± 0:27	16	12:03 ± 0:18	0.0003
	6	15	10:32 ± 0:26	16	12:03 ± 0:18	0.0003
	7	13	10:32 ± 0:28	15	12:04 ± 0:19	0.0005
Left	1	19	10:34 ± 0:30	19	12:03 ± 0:17	0.0001
	2	18	10:31 ± 0:28	18	12:01 ± 0:16	0.0002
	3	18	10:33 ± 0:27	18	12:01 ± 0:16	0.0001
	4	17	10:31 ± 0:28	17	12:01 ± 0:16	0.0009
	5	15	10:36 ± 0:23	16	12:01 ± 0:17	0.0002
	6	15	10:28 ± 0:28	16	12:00 ± 0:18	0.0005
	7	13	10:25 ± 0:29	15	11:58 ± 0:23	0.0005

Note: *p* < 0.000001 after Bonferroni correction for each comparison of mean values before and after CMR therapy using paired *t*-tests.

## Data Availability

Data are available upon request.

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
