# Peer review of "Clinical and MRI-Based Assessment of Patients with Temporomandibular Disorders Treated by Controlled Mandibular Repositioning"

_diagnostics, 2024, doi:10.3390/diagnostics14060572_

Round 1
Reviewer 1 Report
Comments and Suggestions for Authors
The described paper show a interesting approach in therapeutic position using the controlled mandibular repositioning reducing the anteriorly displaced discs. The results are supported by statistically significative studies according to Rm results.
We agree for publishing the submitted paper
Author Response
Dear Sir/Madam
First of all thank you very much for accepting to be a reviewer of our work.
I am feeling grateful for the comments and your valuable feedback which will motivate students like me improve more and more in field to Dentistry.
Kind Regards
Diwakar Singh and greetings from my mentors.
Reviewer 2 Report
Comments and Suggestions for Authors
The paper written by singh et al. is an interesting analysis on the use of the controlled mandibular repositioning via clinical evaluation and MRI exam. The paper is really interesting and well written but there are some issues that could be improved before being published.
The research field of this paper has reached a big interest in the past few years, but the authors cited only references that seem a bit outdated.
The reference format is not the one suggested in the authors guidelines provided by the editor.
The authors analyzed 20 patients and 37 joints, but the author did not mention if there could be differences between patients with bilateral and not TMD. Authors should highlight or justify this difference.
Author Response
Dear Sir/Mam,
Thanks for you valuable feedback, it was a great learning experience for me and i have done all the suggested and highlighted in yellow. Sir/Mam there a attachment regarding your feedback hopefully i have answered them well.
Once again thanking you
Kind regards
Diwakar Singh
